# Effect of Decreasing Temperature Reciprocating Upsetting-Extrusion on Microstructure and Mechanical Properties of Mg-Gd-Y-Zr Alloy

**Wenlong Xu [1], Jianmin Yu [1],*, Guoqin Wu [1], Leichen Jia [1], Zhi Gao [1], Zhan Miao [1], Zhimin Zhang [1] and Feng Yan [2]**

[1] School of Material Science and Engineering, North University of China, Taiyuan 030051, China; wenlong0610@hotmail.com (W.X.); wuguoqin5656@hotmail.com (G.W.); jlc226688@hotmail.com (L.J.); gaozhi000@126.com (Z.G.); niukingway@163.com (Z.M.); nuczhangzhimin@163.com (Z.Z.)

[2] Xi'an Modern Chemistry Research Institute, Xi'an 710065, China; yanfeng259@163.com

* Correspondence: yujianmin@nuc.edu.cn; Tel.: +86-139-3422-4122

**Abstract:** The decreasing temperature reciprocating upsetting-extrusion (RUE) deformation experiment was carried out on Mg-Gd-Y-Zr alloy to study RUE deformation on the influence of microstructure of the alloy. This work showed that with the gradual increase of RUE deformation passes, the continuous dynamic recrystallization (CDRX) process and the discontinuous dynamic recrystallization (DDRX) process occurred at the same time, and the grain refinement effect was obvious. Particulate precipitation induced the generation of DRX through particle-stimulated nucleation (PSN). In addition, after one pass of RUE deformation, the alloy produced a strong basal texture. As the RUE experiment proceeded, the basal texture intensity decreased. The weakening of the texture was due to the combined effect of DRX and alternating loading forces in the axial and radial directions. After four RUE passes, the mechanical properties of the alloy had been significantly improved, which was the result of the combined effect of dislocation strengthening, fine grain strengthening, and second phase strengthening.

**Keywords:** Mg-Gd-Y-Zr alloy; repetitive upsetting-extrusion; grain refinement; texture; microstructure; mechanical properties

## 1. Introduction

In recent years, Mg alloy has become one of the most important commercial alloys. Because of its low density, high specific strength and good shock absorption performance, Mg alloys are ideal materials for lightweight and energy saving [1–3]. However, the application of Mg alloys is limited because of its low strength and poor machinability, which is as a result of the limited activation ability of non-basal slip systems at room temperature (RT) in the magnesium hexagonal close-packed cell. [4–7]. In recent years, enormous studies have shown that adding appropriate amount of rare earth (RE) elements to Mg alloys can bring about solid solution strengthening and aging precipitation strengthening, which will significantly improve the strength of the alloy [8–12]. The severe plastic deformation (SPD) techniques, for instance equal-diameter angular extrusion, high-pressure torsion, multi-directional forging and cyclic extrusion, etc., can refine grains better than conventional plastic deformation [13–16], while improving the strength, plasticity and toughness of the material.

Repetitive upsetting-extrusion (RUE) is a sort of SPD technique, which combines the advantages of upsetting and extrusion. It can keep the size and cross-sectional area of the blank in the process of deformation, and keep the original size after processing [17]. RUE can be used to deform large-sized billets in engineering, and have broad application prospects in industrial production. Current research

on AZ61, AZ80 and other materials has confirmed that the RUE can refine the grains and the second phase to obtain a more uniform microstructure [18–20]. Cui et al. [21] studied on the slabs of Mg-Gd-Y-Zn-Zr alloy showed that the alloy's RT tensile strength was about 190 MPa, the yield strength was about 140 MPa, and elongation was about 1%, but it had undergone multi-pass plasticity after deformation and aging treatment, RT tensile yield strength (TYS), ≥410 MPa, ultimate tensile strength ≥390 MPa, and elongation ≥7%. Du et al. [22] studied the microstructure evolution of Mg-13Gd-4Y-2Zn-0.5Zr (wt%) in RUE, which drew the grain size was refined to 3.4 µm after six passes of deformation. Zhang et al. [23] studied the microstructure evolution of Mg-RE-Zn alloy by RUE under different decreasing temperatures, and the results showed that the volume fraction of DRX decreased and the average grain size increased with the decrease of deformation temperature.

Among various SPD technologies, RUE effectively eliminates anisotropy and produces ultrafine crystals [22,23]. However, because Mg alloys with high Gd content are expensive, and the addition of Zn element to RE-Mg alloy will produce the long-period stacked ordered (LPSO) phase, which can inhibit the formation of DRX [24]. Therefore, Mg alloys with low Gd content and no addition of Zn have high application prospects. In this paper, we analyzed the effect of the decreasing temperature RUE process on the microstructure and mechanical properties of Mg-8Gd-3Y-0.5Zr (wt%) alloy, and further revealed the grain refinement law under RUE.

## 2. Materials and Methods

The Mg-8Gd-3Y-0.5Zr alloy made by semi-continuous casting was homogenized at 520 °C for 16 h, cooled in air, and processed into a cylindrical specimen of Φ50 mm × 230 mm. Figure 1a shows the working principle of RUE. First of all, upsetting treatment was performed to the alloy, after the upsetting treatment the diameter of the alloy became D = 70 mm. Afterwards proceed to the next extruded operation. The specification of the extruded material was d = 50 mm, and a round of RUE experiment operation was accomplished. Repeated the above operation process, until the four passes of deformation were completed.

The RUE deformation experiment was implemented on a 630 KN press with a strain rate of $0.002 \text{ s}^{-1}$. The schematic diagram of the experimental process is shown in Figure 1b. The RUE temperature is reduced from 420 °C to 390 °C, each pass is reduced by 10 °C, and the formula is $\varepsilon = 4n \ln D/d$ [25], the cumulative strain is calculated to be about 5.38, and the cumulative strain of each RUE pass (n = 1) is about 1.345. Before the experiment, in order to make sure the mold and alloy were kept at the same temperature throughout the entire deformation process, it was necessary to heat the mold and the material to more than 30 °C as well as the temperature for 2 h at the same time, and then lubricated with oil-based graphite lubricant. The experiment was carried out in four passes in total. After each pass, the sample was placed in the air and gradually cooled to RT.

In order to make sure the uniformity of the experimental conditions, the sampling position of the metallographic samples used for microscopic analysis was at the center of the extruded material (Point A position as shown in Figure 1). The surface parallel to the extrusion direction (ED) was selected for microstructure analysis. The sandpaper was polished and subjected to conventional mechanical polishing, then etched with acetic acid picric acid etchant (1 g picric acid, 2 mL acetic acid, 2 mL distilled water and 14 mL alcohol) to observe the optical microstructure. The microstructure of the alloy was observed using an optical microscope (OM; DM2500M, Leica Microsystems, Wetzlar, Germany), scanning electron microscope (SEM; SU5000, Hitachi, Tokyo, Japan), and electron back-scatter diffraction (EBSD, EDAX Inc., Mahwah, NJ, USA). The composition of the phase in the alloy was measured and analyzed by X-ray diffraction (XRD; Rigaku D/MAX2500PC, Rigaku, Tokyo, Japan). XRD analysis was performed on samples of each RUE pass using Smart-Lab, and the diffraction angle was 20–90°. X-ray energy spectroscopy experiments were performed at 50 KV, and the step size was 0.01. The EBSD analysis method was used to characterize the grain size and texture of each RUE pass, and the data were analyzed using Orientation imaging microscopy (OIM) analysis software (EDAX Inc., Mahwah, NJ, USA). In order to make sure the accuracy of the data, more than 300,000 grains were selected

for analysis in each sample. In order to determine the mechanical properties of the alloy after RUE deformation, the initial state alloy and different RUE passes alloy were processed into dog bone shaped along the parallel extrusion direction (sample size was shown in Figure 1c). Obtained by using Instron 3382 universal testing machine (INSTRON, Norwood, MA, USA) at RT at a tensile rate of 0.01 mm/min.

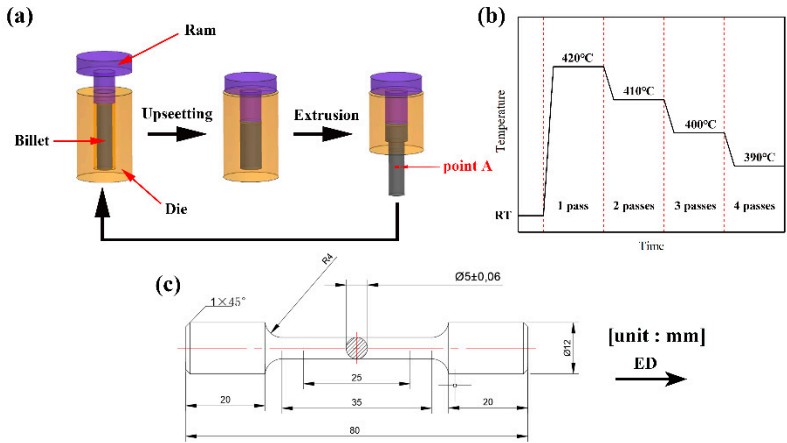

**Figure 1.** (**a**) Work functional picture of RUE; (**b**) Flow diagram of RUE experiment; (**c**) Dimensions of specimens for tensile test.

## 3. Results and Discussion

### 3.1. Microstructure Evolution of the RUE Alloys

Figure 2 shows the microstructure of the initial state alloy. Micrographs are longitudinal section, and the grain distribution showing equiaxed grains as a whole. The initial state alloy grains are relatively coarse, the average grain size is 34.8 μm, and it can be observed that there are particle phases precipitated inside the alloy (shown by the red arrows in Figure 2).

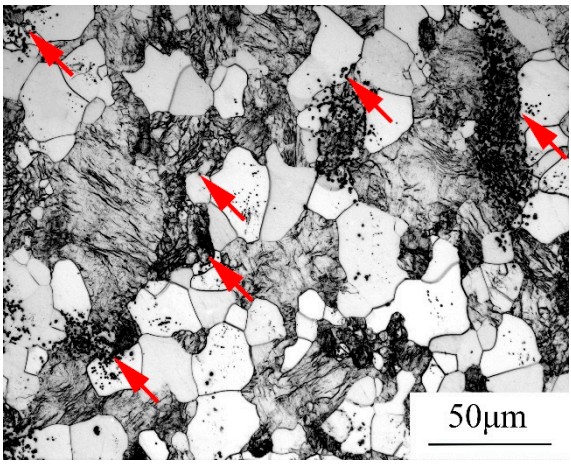

**Figure 2.** Optical image of initial state alloy.

Figure 3 shows the microstructure of each RUE pass taken along the longitudinal section. After one pass of RUE deformation, the alloy exhibited a typical bimodal microstructure. The coarse primary grains were elongated along the deformation direction, and small grains distributed in a chain shape appeared at the tridentate grain boundaries of the coarse grains. This indicated that DRX occurs during the deformation process. It was due to the stress concentration at the pristine grain boundary, severe deformation and high dislocation density. In the RUE stage, the accumulated dislocations near the coarse grain boundary were rearranged to form a low angle grain boundary, which resulted

in the formation of sub-grains or sub-structures [26]. At the same time, some fine particle phases precipitated at the DRX grain boundaries, which indicated that the alloy had undergone dynamic precipitation during the RUE deformation process. The slippage of the particles relative to the grain boundaries plays a pinning role, inhibiting the growth of dynamic recrystallized grains during thermal deformation and during heating between passes [24]. With the increasing number of the RUE passes, the size of the remaining coarse grains gradually decreases, and the number of DRX grains continues to increase, surrounding the coarse grains. At this time, the grain size distribution is still uneven, and the volume fraction of large grains is relatively large. After four passes, the grain of the alloy is significantly refined and uniformly refined, and the grain boundary blur in the DRX region is difficult to distinguish.

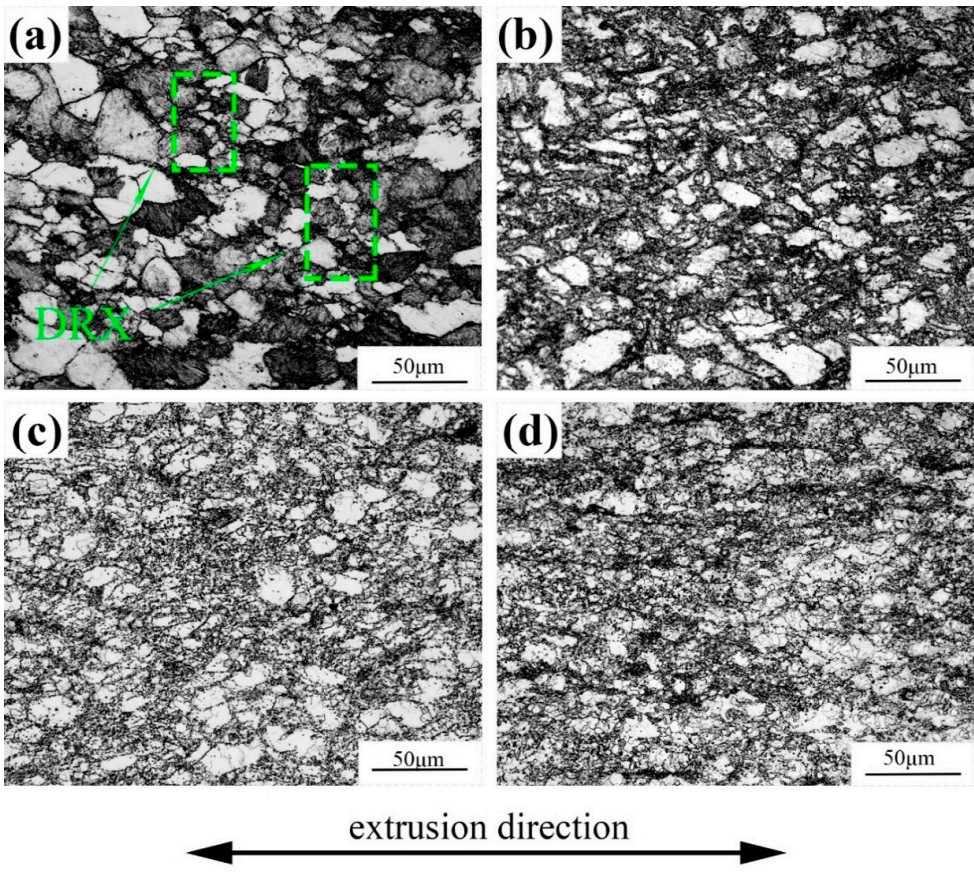

**Figure 3.** Optical images of Mg-Gd-Y-Zr RUE alloys: (**a**) one pass; (**b**) two passes; (**c**) three passes; (**d**) four passes. Micrographs are longitudinal section.

Figure 4 shows the SEM-BSE (back-scattered electron) images, XRD diffraction analysis, and EDS elemental analysis results of each RUE pass taken along the longitudinal section. According to XRD diffraction analysis results (Figure 4e) and EDS element analysis results (Figure 4f) for point B, the alloy after multiple passes of RUE deformation, it is principally consisted of $\alpha$-Mg matrix phase and Mg5 (Gd, Y) phase [26], while no new phase is formed. As the RUE deformed to four passes, the Mg5 (Gd, Y) phase is uniformly dispersed. It can be seen in the Figure 3a that the Mg5 (Gd, Y) phase mainly precipitates at the DRX grain boundaries. This is because a large number of defects are distributed in these regions, which is conducive to the diffusion of atoms. PSN accelerates the formation of DRX, which indicates that strain-induced precipitation is the main mechanism of Mg5 (Gd, Y) phase [27]. The SEM images of different RUE deformation passes were binarized. The results showed that with the increasing RUE deformation passes, the content of Mg5 (Gd, Y) phase increased. This is because the deformation temperature continues to decrease, the solid solubility of rare earth atoms in the alloy

will gradually decrease with the reduction of temperature, and the atomic diffusion will be faster in the region with higher strain at the grain boundary, which is beneficial to Mg5 (Gd, Y) Phase precipitation.

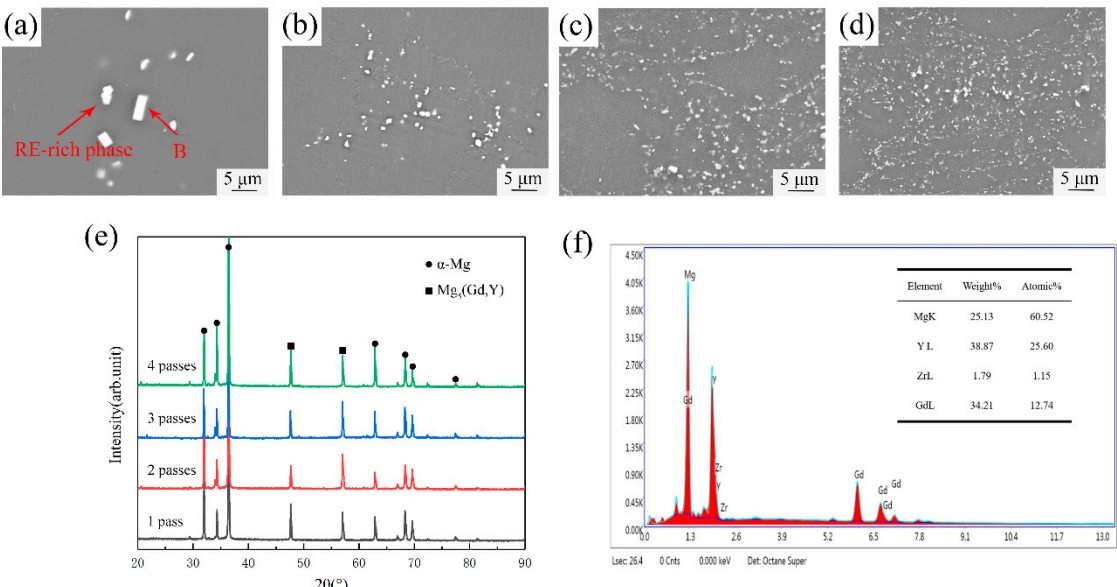

**Figure 4.** SEM-BSE images of alloys under different RUE passes: (**a**) one pass; (**b**) two passes; (**c**) three passes; (**d**) four passes. Micrographs are longitudinal section. (**e**) XRD patterns of the alloys under different RUE passes. (**f**) EDS elemental analysis of the phase marked B in Figure 4a.

After four RUE passes, the grains were significantly refined, and the grain boundaries became blurred. For the sake of further characterize the grain size change during the decreasing temperature RUE process, each RUE deformation pass was analyzed and characterized by EBSD.

Figure 5 shows grain size distribution of all grains and DRX grains of the alloy under different RUE passes, and it shows the DRX grains respectively. Different colors represent different grain orientations, red for <0001>, blue for <10-10>, and green for <2-1-10>. The black area in the picture has a low confidence factor (CI < 0.1), and no Kikuchi pattern map is obtained. Therefore, the area which cannot be calibrated may be the precipitated Mg5 (Gd, Y) particle phase, because the phase information is not included in the database. After one pass of RUE deformation, the alloy is principally consisted of coarse original grains and tiny DRX grains generated at the grain boundaries. The coarse primary grains are mainly green and pink, which manifest that the grains present disparate orientations, and color changes are found inside the grains, indicating that there is also lattice rotation in the grains due to deformation [28]. After four passes of RUE deformation, the alloy structure is completely composed of fine dynamic recrystallization, showing random orientation distribution characteristics, and ultrafine grain regions with grain sizes less than 1 μm are found in the fine grain regions.

At the same time, the grain size distribution after different RUE deformation passes is also shown in Figure 5. In order to characterize the grain size change during the RUE deformation process in cooling, grains with a grain size of less than 10 μm were defined as fine grains. It can be seen from the Figure 5i–l that the grain refinement of the alloy mainly occurs in one pass and two passes, and the average grain size is reduced from 32.7 μm to 8.96 μm, reduced by nearly 72.6%. At this time, the average grain size has been lessened. It is close to the standard of fine grains, and with the number of RUE passes further increases, the grain refining effect weakens. After four passes of deformation completed, a relatively uniform fine-grained structure is obtained, and the average grain size is refined to 6.72 μm. This may be due to the fact that the dynamic recrystallization grains occupy most of the area, which can better coordinate the further deformation and inhibit the dislocation proliferation. When the dislocation density is lower than the need of the sub crystal nucleation, it becomes very difficult to refine the grains further through the dynamic recrystallization. The volume fraction of

recrystallization and grain size show the same law. With the increasing number of RUE deformation passes, the original coarse grain area is gradually replaced by the fine dynamic recrystallization area. After four passes were completed, the volume fraction of DRX reached 86.7%.

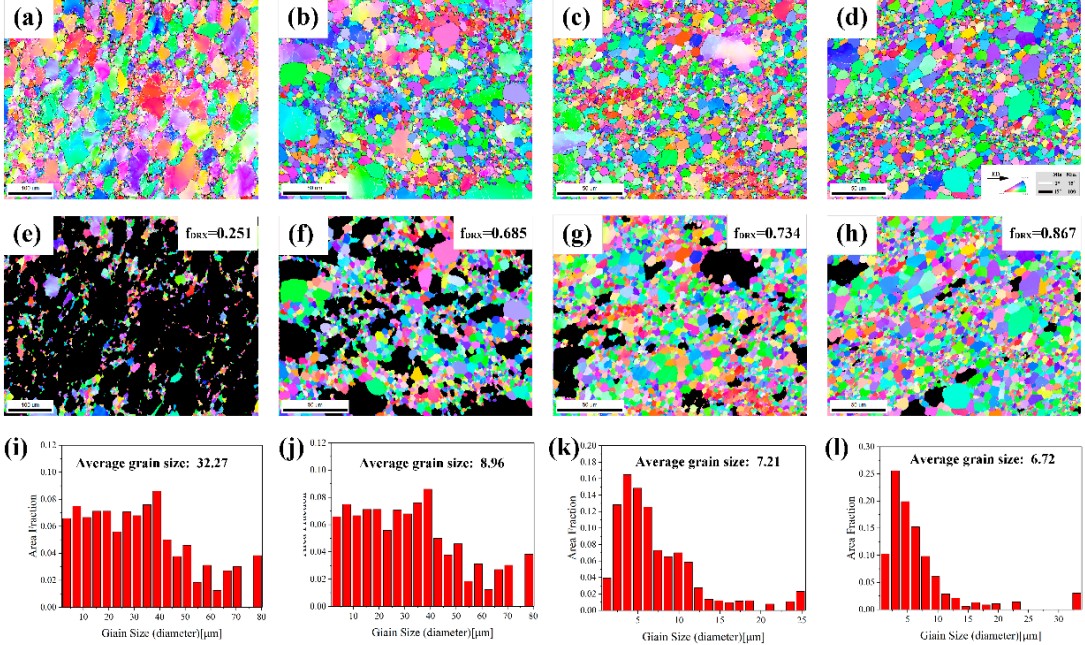

**Figure 5.** OIM maps and grain size distribution of all grains and DRX grains of the alloy under different RUE passes. (**a**,**e**,**i**) one pass; (**b**,**f**,**j**) two passes; (**c**,**g**,**k**) three passes; (**d**,**h**,**l**) four passes.

Figure 6 shows the angular distribution of grain boundary orientations and misorientation angle distribution through different RUE deformation passes. Figure 6a,e shows that after the RUE deformation of one pass, most of the large-angle grain boundaries are primarily distributed at the grain boundaries of bulky grains, and a small amount is distributed at the newly generated DRX grain boundaries, while the small angular grain boundaries are mainly distributed near the grain boundaries inside the coarse grains. At this time, a clear distribution peak of the low angel boundaries (LABs) is 5–15 degrees. Because of the small amount of deformation, a large amount of coarse primary crystallization and a small amount of DRX are the main microstructure characteristics. It is mainly DDRX at this time. With the increasing number of RUE passes, the continuous accumulation of strain leads to the increase of dislocations in the grain, dislocation movement, dislocation reaction, and formation of sub-grains composed of small-angle grain boundaries. Finally, it absorbs dislocations and transforms into recrystallized Large-angle grain boundaries. This shows that CDRX occurs during the RUE process at a reduced temperature, and the LABs gradually transform into high angle boundaries (HABs), forming new grains. CDRX is an important nucleation mechanism in Mg alloys, which involves the nucleation and growth of sub-grains [29,30]. Vogel et al. reported that in the early stage of hot deformation, the slip in the shear deformation area, especially the basal slip, which caused the internal dislocations of the grain to increase continuously. The cell structure was formed by the dislocation entanglement, which would be flattened to form sub-crystals, and then the new grains were formed through the migration and merge of sub-grain boundaries (see Figure 7) [31]. However, both mechanisms rely on the coarsening of sub-grains to develop into DRX cores, whether through the merging or migration of sub-grains to form new grains. The sub-grains were formed by multilateralization under the severe strain conditions, which grow into an effective core of DRX by consuming the surrounding high energy regions. Therefore, as the cumulative strain increased, more sub-grains would be produced, which was conducive to recrystallization nucleation. After four passes of RUE deformation (Figure 6d, h), most of the alloy area was occupied by fine DRX grains, while a

large number of small angles can still be seen inside the recrystallized grain boundary distribution, which indicates that the DRX grains are also gradually refined.

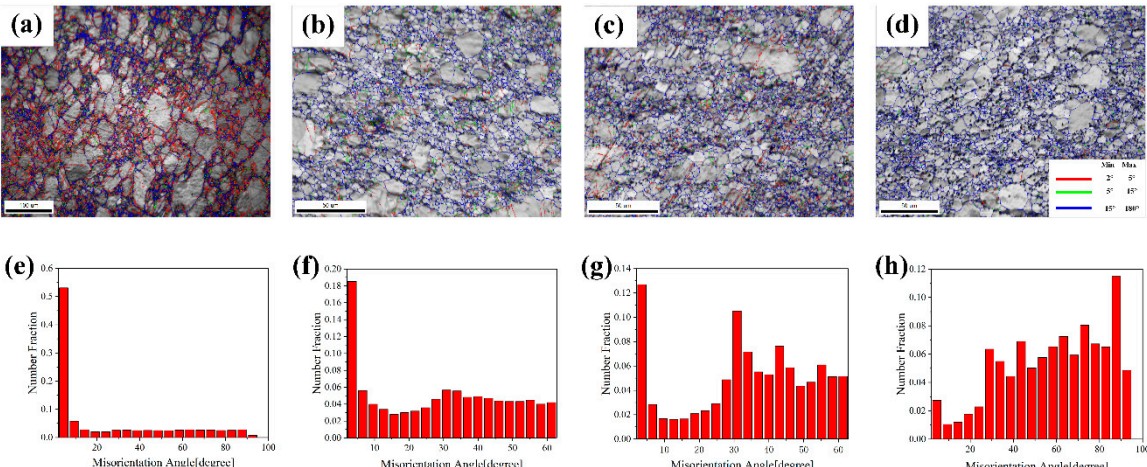

**Figure 6.** Grain boundary structure distribution diagram and misorientation angle distribution of the alloy under different RUE passes. (**a**,**e**) one pass; (**b**,**f**) two passes; (**c**,**g**) three passes; (**d**,**h**) four passes.

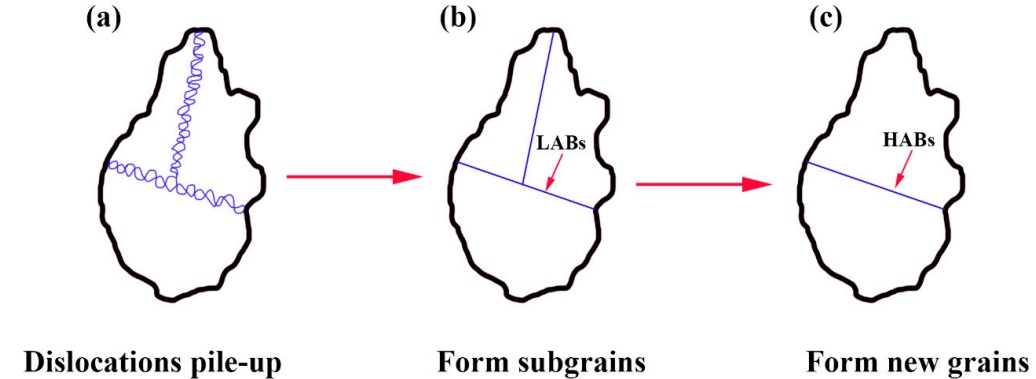

**Figure 7.** Schematic diagram s of CDRX of Mg alloys. (**a**) Dislocations pile-up interact to form cell-like substructure; (**b**) Dynamic recovery to form sub-grains; (**c**) Sub-grain boundaries migrate and merge to form new grains with HABs.

### 3.2. Texture Evolution of the RUE Alloys

For the sake of further elucidate the texture evolution during RUE process, the EBSD was used to analyze and characterize each RUE deformation pass.

Figure 8a–l shows the (0001) pole figures of the DRX grains, UNDRX grains, all grains and the (0001) inverse pole figures of all grains. It can be displayed that after one pass of RUE deformation, a typical basal texture appears in the alloy. A large number of studies have shown that the fiber texture of Mg alloys exhibits basal plane orientation characteristics during extrusion and compression deformation, that is, the (0001) basal plane and <10-10> crystal orientation are parallel to the extrusion direction during extrusion. The (0001) base plane are perpendicular to the direction of the compression axis [32–37]. After the alloy undergoes RUE deformation, the extreme density region of the (0001) base plane is perpendicular to the extrusion direction. Lin et al. [38] reported this type of texture. This is because during the deformation process, the loading direction is alternated between axial and radial directions, and most of the (0001) bases are deflected toward the direction of extrusion. With the increasing number of the passes of deformation, the (0001) base surface exhibits a tendency of periodic rotation with alternating changes in loading direction. Therefore, after one pass of RUE deformation of the alloy, base texture is gradually weakened, and the strength of the base surface is

more dispersed. As shown in Figure 8i,l, as the deformation passes increase from one pass to four passes, the maximum texture strength pole density decreases from 5.866 to 2.873. At the same time, because the alloy generates a large number of DRX grains when it undergoes hot deformation, these grains show a random grain orientation, and the texture intensity of the DRX grains becomes increased (see Figure 8a–d). As the degree of cumulative strain increases, the volume fraction of the DRX grains increased (see Figure 8e–h), which to some extent offsets the deformation texture of the non-dynamic recrystallization (UNDRX) grains. With the increasing number of passes, the alloy will precipitate a large amount of Mg5 (Gd, Y) particle phases at the grain boundaries. The particles also have a certain hindering effect on the rotation of the crystal lattice, which has a deformed texture on the alloy, causing a significant weakening effect.

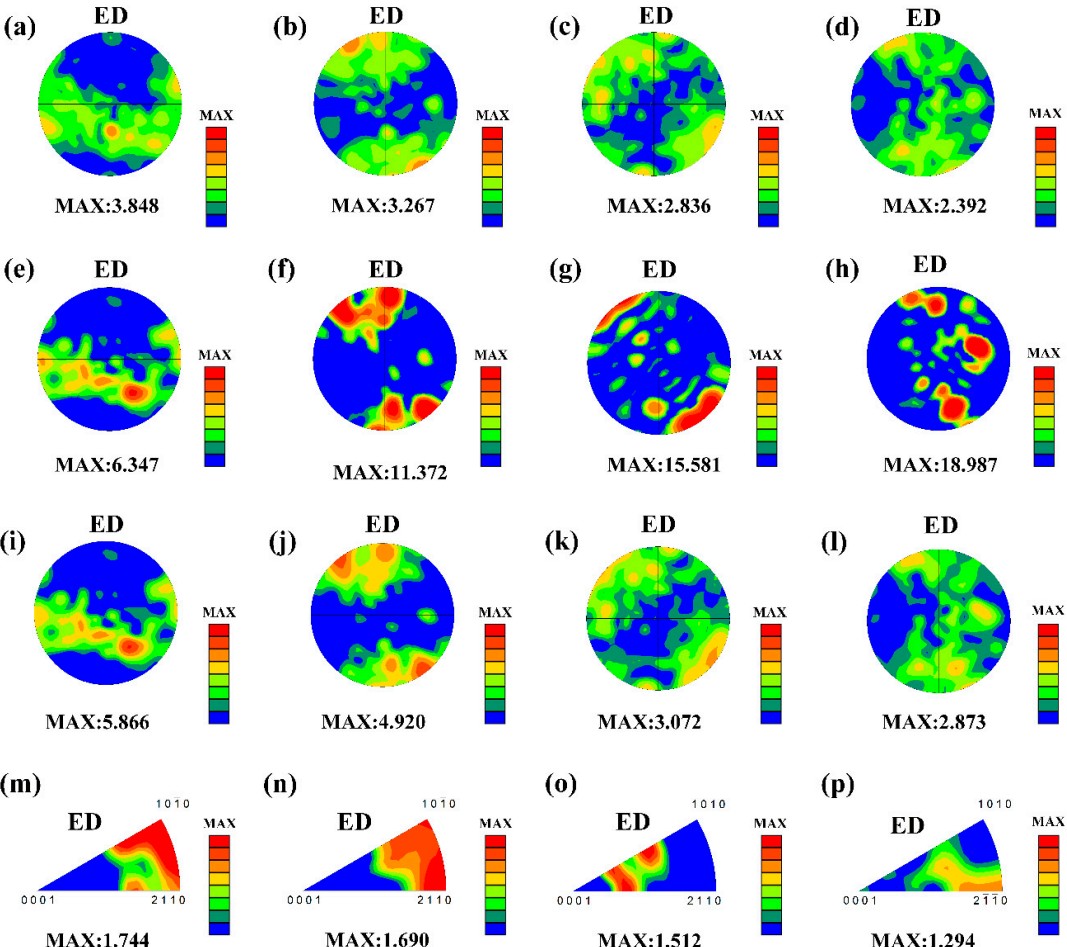

**Figure 8.** (**a**–**d**) (0001) Pole figures of DRX grains; (**e**–**h**) (0001) pole figures of UNDRX grains; (**i**–**l**) (0001) pole figures of all grains; (**m**–**p**) (0001) inverse pole figures of all grains; (**a**,**e**,**i**,**m**) one pass; (**b**,**f**,**j**,**n**) two passes; (**c**,**g**,**k**,**o**) three passes; (**d**,**h**,**l**,**p**) four passes.

### 3.3. Mechanical Properties of the RUE Alloys

Figure 9 shows the mechanical properties of initial state and different RUE passes Mg-8Gd-3Y-2Zn-0.5Zr alloy at RT. It can be seen that during the RUE deformation of the alloy, the ultimate tensile strength (UTS), tensile yield strength (TYS) and elongation of the alloy show an upward trend with the increased of RUE passes. After four passes, the UTS, TYS and elongation of the alloy reached 320 ± 3.2 MPa, 258 ± 2.6 MPa and 11.7 ± 0.1%, respectively. The standard deviation of the machine is 1%. Meanwhile, during the process of RUE deformation, a large number of dislocations

would emerge inside the grains, resulting in slip and dislocations entanglement, which would cause the grains to be elongated and broken. According to the Hall–Petch relationship [39]:

$$\sigma_s = \sigma_0 + kd^{-1/2} \tag{1}$$

Among them, $\sigma$ is the TYS of the material, $\sigma_0$ is constant, $d$ is the average grain size of the alloy. It can be seen that the smaller the average grain size, the greater the yield strength of the alloy. The ductility of the material increases with the increase of RUE passes, which can be attributed to the fact that the RUE process will significantly refine the grains, and a large number of Mg5 (Gd, Y) phases are precipitated at grain boundaries and dispersed uniformly in the alloy, which encountered dislocation motion and improved the mechanical properties of the alloy. Grain refinement can simultaneously improve the strength and ductility of metallic materials. At the same time, owing to the effect of DRX during the RUE process, the grain orientation changed, and the basal texture intensity decreased, which made the material anisotropy smaller and the ductility improved. [40]. Therefore, the mechanical properties of the alloy were improved by the joint action of dislocation strengthening, fine crystal strengthening and second-phase strengthening.

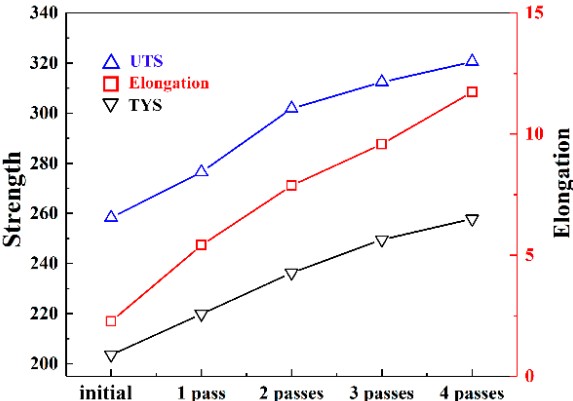

**Figure 9.** Mechanical properties of initial state and different RUE passes alloys.

## 4. Conclusions

This article systematically studies the changes of microstructure of Mg-8Gd-3Y-0.5Zr alloy after decreasing temperature RUE deformation. The following results are drawn:

(1) The alloy exhibits a typical bimodal structure in the RUE process. The grains are gradually refined, and after four passes of deformations, the average grain size reaches 6.72 μm.

(2) After four passes of RUE deformation, the alloy is principally consisted of $\alpha$-Mg matrix phase and $Mg_5$ (Gd, Y) phase. A large number of $Mg_5$ (Gd, Y) phase precipitated along the DRX grain boundary, and the PSN accelerates the formation of DRX.

(3) As the RUE experiment proceeds, the basal texture intensity decreases, which is due to the DRX grains, diffused Mg5 (Gd, Y) phase and the alternating axial and radial transformation of the loading direction during the deformation process. The basal strength distribution is scattered. Most (0001) bases are deflected toward the direction of the squeeze.

(4) After 4 RUE passes, the alloy has excellent mechanical properties, UTS is 320 ± 3.2 MPa, TYS is 258 ± 2.6 MPa, and the elongation is 11.7 ± 0.1%, which is the result of dislocation strengthening, grain refinement, and Mg5 (Gd, Y) phase dispersed distribution.

**Author Contributions:** J.Y. and G.W. designed the experiments; L.J., Z.G., and Z.M. conducted the experiments and collected the data; Z.Z. and F.Y. analyzed the data; W.X. wrote the paper. All authors have read and agreed to the published version of the manuscript.

**Funding:** This research was financially supported by the Natural Science Foundation of Shanxi Province (No.201901D111176) and Key R&D Program of Shanxi Province (International Cooperation) (No.201903D421036) and Scientific and Technological Innovation Programs of Higher Education Institutions in Shanxi (No.2018002).

**Conflicts of Interest:** The authors declare no conflict of interest.

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
