# Peer review of "Effect of Decreasing Temperature Reciprocating Upsetting-Extrusion on Microstructure and Mechanical Properties of Mg-Gd-Y-Zr Alloy"

_metals, doi:10.3390/met10070985_

Round 1

Reviewer 1 Report

The paper deals with microstructure and mechanical properties of the complex Mg-Gd-Y-Zr alloy. The material was prepared by reciprocating upsetting extrusion at decreasing processing temperature. The authors documented very thoroughly the microstructure and texture of processed samples. On the other hand, the paper has only a descriptive character without any attempt to interpret experimental results.

Referee’s remarks:

  1. Line 33: instead of…..result of the limited activation ability of the slip system of hexagonal dense packed lattice structure…better: …result  of the limited activation ability of non-basal slip systems at room  temperature in the magnesium hexagonal close packed cell.
  2. The basal plane is in the Mg cell only one, then better (0001) instead of .
  3. Line 188: Usually as low angel grain boundaries are depicted grain boundaries with the disorientation 5-15 degreases.
  4. It would be better to introduce the strain rate instead of the machine speed.
  5. The accuracy of the stress value on two decimal places is not possible. State the standard deviation!!
  6. Explain meaning of s0, M, a, G, b, p in eq. 1.
  7. In the eq. 1 two strengthening terms are taken into account. I doubt if they may be simply summed. Could you assume that the dislocation density is only increasing? It would be interesting to show the Hall-Petch dependence.
  8. The material ductility was found to be increasing with increasing number of processing passes. Have authors some explanation for this finding?
  9. Authors discussed results only qualitatively. Deeper inside to deformation mechanisms is missing.  Discuss influence of twinning during the tensile testing.
  10. English of the paper is acceptable.

I recommend major revision of the paper with the emphasis on the interpretation of experimental results.

Reviewer 2 Report

In the paper entitled “Effect of decreasing temperature reciprocating upsetting-extrusion on microstructure and mechanical properties of Mg-Gd-Y-Zr alloy”, Authors present the results of studies which concerns the describing of the effects of continuous dynamic recrystallization (CDRX) process and the discontinuous dynamic recrystallization (DDRX) process occurred during reciprocating upsetting-extrusion (RUE) of Mg-Gd-Y-Zr alloy. The title and abstract are appropriate for the content of the text. Furthermore, the article is well constructed, the experiments were well conducted, and analysis was well performed. The manuscript gives adequate contribution in the investigated field and it can be accepted for publications after minor revision. However, I have a few comments which may be allowed to improve the quality of your work. The details are attached below.

1) Line 85 – Please answer why for microstructure analysis was selected the surface parallel to the extrusion direction (ED) instead of the surface perpendicular to ED?

2) Line 92 – Give more details about XRD analysis, X-ray lamp type (wavelength), step size, time per step (counting time) as well as name and version of crystal structure database.

3) Line 102 – Please add a short description of the microstructure of the as-cast alloy and mark phases in figure 2.

4) XRD diffraction analysis –  Valuable information would add a calculation (based on XRD patterns) of average crystallite size and dislocation density. This accumulated energy of plastic deformation (in the form of a dislocation substructure) leads to a decrease in the recrystallization temperature.

Round 2

Reviewer 1 Report

The authors address acceptably all comments. Paper can be published.

Author Response

Thank you for your comments. Those comments are all valuable and very helpful for revising and improving our paper, as well as the important guiding significance to our researches.